# FirebotSLAM: Thermal SLAM to Increase Situational Awareness in Smoke-Filled Environments

**DOI:** 10.3390/s23177611

**Published:** 2023-09-02

**Authors:** Benjamin Ronald van Manen, Victor Sluiter, Abeje Yenehun Mersha

**Affiliations:** Smart Mechatronics And RoboTics (SMART) Research Group, Saxion University of Applied Sciences, Ariënsplein 1-300, 7511 JX Enschede, The Netherlands; v.i.sluiter@saxion.nl (V.S.); a.y.mersha@saxion.nl (A.Y.M.)

**Keywords:** search and rescue, hazardous environment, SLAM, computer vision, thermal

## Abstract

Operating in extreme environments is often challenging due to the lack of perceptual knowledge. During fire incidents in large buildings, the extreme levels of smoke can seriously impede a firefighter’s vision, potentially leading to severe material damage and loss of life. To increase the safety of firefighters, research is conducted in collaboration with Dutch fire departments into the usability of Unmanned Ground Vehicles to increase situational awareness in hazardous environments. This paper proposes FirebotSLAM, the first algorithm capable of coherently computing a robot’s odometry while creating a comprehensible 3D map solely using the information extracted from thermal images. The literature showed that the most challenging aspect of thermal Simultaneous Localization and Mapping (SLAM) is the extraction of robust features in thermal images. Therefore, a practical benchmark of feature extraction and description methods was performed on datasets recorded during a fire incident. The best-performing combination of extractor and descriptor is then implemented into a state-of-the-art visual SLAM algorithm. As a result, FirebotSLAM is the first thermal odometry algorithm able to perform global trajectory optimization by detecting loop closures. Finally, FirebotSLAM is the first thermal SLAM algorithm to be tested in a fiery environment to validate its applicability in an operational scenario.

## 1. Introduction

Missions in extreme situations require acute knowledge of the environment to be accomplished safely and successfully. Fires in large buildings, for example, are life-threatening situations for firefighters. During an incident inside a parking garage, a burning car can rapidly produce enough smoke to make it difficult to find the source of the fire or to navigate towards an exit. In industrial buildings, toxic chemicals may additionally have been released into the air. For these reasons, it is sometimes too dangerous for firefighters to enter the building to fight the fire, which may result in loss of life and severe material damage, as shown in Figure 1.

In collaboration with Dutch fire departments, research has been conducted into the use of Unmanned Ground Vehicles in order to improve situational awareness of firefighters during fire incidents. Preliminary experiments in daily practice, however, showed that their usability proved to be restricted. The main deficiency of these vehicles is namely the limited field of view from the on-board cameras. Furthermore, the “tunnel vision” effect generated by the video feeds alone does not provide the desired general overview of the incident. Hence, the need for increased situational awareness in the form of a 3D map arose.

Contrary to the popular sensors used to perform Simultaneous Localization and Mapping (SLAM), such as visual cameras and LiDAR, thermal cameras have the capability to perceive a smoke-filled environment. Although thermal SLAM is an image-based SLAM method, obtaining a 3D map from thermal images is not as straightforward as providing the images to a state-of-the-art visual SLAM algorithm. Thermal cameras namely possess some additional challenges, such as a low resolution, a low signal-to-noise ratio worsening over time and low contrast [1], as can be observed in Figure 2. Additionally, raw radiometric data from thermal cameras is 16-bit, whereas common computer vision algorithms only function using 8-bit images, resulting in data loss in environments with highly varying thermal gradients. The presence of smoke and fire may also affect the quality of the thermal images. Thicker smoke particles may still reflect some LWIR rays “fogging” the image, thus making feature tracking more difficult. Finally, fire is dynamic, which potentially introduces some error in the pose estimation. The latter namely assumes a static environment.

In this paper, the feasibility of performing thermal SLAM during a fire incident to obtain greater situational awareness is assessed. As a result, the following contributions are made by the authors:A practical benchmark of feature extraction and description methods on thermal images recorded during a fire incident.The first thermal SLAM algorithm capable of creating a comprehensible 3D map inside a fiery environment using solely 3D information extracted from thermal images was developed.The first thermal SLAM system that is capable of obtaining a coherent 3D map by performing a global trajectory optimization using bundle adjustment from thermal loop closure detections.The first thermal odometry and mapping algorithm to be tested in a fiery environment to validate the applicability of the proposed algorithm in an operational environment.

Using FirebotSLAM, firefighters are now able to obtain an overview of the fire incident before entering the smoke-filled environment. Although FirebotSLAM does not provide a detailed description of the environment, its generated map includes a general floor plan with the location of relevant obstacles. This floor plan is vital to firefighters to navigate effectively during the fire incident and allows them to execute their rescue operations in a safer and more efficient manner. A video of the obtained results can be found here: https://youtu.be/9pRakFnuySc (accessed on 20 August 2023).

The remainder of this article is structured as follows. Section 2 presents an overview of related work. Section 3 describes the methodology and materials used in this research. The first practical benchmark of thermal feature extraction and description algorithms was performed on datasets recorded during a fire incident. The chosen feature extractor and descriptor are then implemented into a state-of-the-art visual SLAM algorithm. Section 4 presents the results obtained from the benchmark and FirebotSLAM. Finally, Section 5 and Section 6, respectively, provide some overall discussion points on the obtained results and general conclusions reached during this research.

## 2. Related Work

Thermal SLAM is a rather new research topic, with most results being published in the last decade. Research is mainly focusing on thermal odometry rather than thermal SLAM, with several research groups that succeeded in utilizing thermal cameras to compute the odometry of a vehicle. For thermal mapping, i.e., the visualization of thermal data in 3D, the most common method is to superimpose the thermal images onto a pointcloud obtained from data generated from another depth source.

### 2.1. Thermal Feature Extraction and Matching

In a collaboration between the Royal Institute of Technology (KTH) in Sweden and FLIR Systems AB, J. Johansson et al. [2] evaluated the performance of different visual-based feature detectors and descriptors on thermal images. The dataset is composed of individual image pairs taken in benign structural and textured environments. Before evaluation, the images were rescaled in an 8-bit manner and their histograms were equalized. The different algorithms were tested on different image deformations, such as viewpoint change, rotation, scale, noise and downsampling. The different combinations of candidates are evaluated based on the recall, i.e., the number of correct feature matches according to RANSAC with respect to the total amount of matches. Among the floating-point descriptors, the Hessian-affine extractor in combination with the LIOP descriptor showed consistent good performance on all the different image deformations. The SURF extractor and descriptor followed behind, demonstrating particularly good invariance against Gaussian blur and Gaussian white noise. In contrast to working with visual images, binary descriptors offer a similar performance to floating-point descriptors in thermal images at a lower computational cost. Especially, the combination ORB with FREAK or BRISK provided overall competing results.

A second benchmark of feature extraction and description methods of thermal features was performed by T. Mouats et al. [3]. As the targeted application was estimating the odometry of an Unmanned Aerial Vehicle (UAV) using thermal images, the processing time was also considered in the evaluation. The datasets used in the benchmark consisted of a number of video sequences recorded in benign indoor and outdoor situations using a FLIR Tau2 LWIR camera. It was concluded that, in general, blob detectors such as SIFT and SURF provided lower repeatability than corner detectors like FAST and GFTT. This means that, when an object is observed from different perspectives, the likelihood is smaller that features are located at the same location on the object. On the other hand, blob features showed more distinctiveness than corner features as the comparison of their descriptor yielded higher matching scores, i.e., more correct matches. SIFT feature extraction performed the worst in almost all aspects. Feature extractors seem not to be affected by motion blur, with the SURF blob extractor performing the best. It was observed that the descriptors were less influenced by non-uniformity noise (NUC) than expected. The effect seemed, however, to be stronger in indoor environments, where objects often have a uniform temperature, providing less contrast to the image and thus amplifying the noise visibility. The authors concluded by proposing SURF feature extraction in combination with the FREAK binary descriptor as the best combination for thermal navigation applications. The proposed combination offers a good tradeoff between good matchability as well as repeatability while also providing a satisfactory computation time for real-time navigation.

The two benchmarks presented here are performed using datasets recorded in benign environments with a uniform temperature. Both smoke and intense heat sources have a substantial impact on the environment, which may affect the feature extraction and description process as well. Furthermore, a uniform distribution of the features is desired for a more accurate estimation of the odometry [4]. Notably, in the benchmark from Mouats et al. [3], which intends to apply feature extraction and description in thermal odometry estimation, a metric capable of quantifying the feature distribution across the frame is lacking.

### 2.2. Thermal Odometry

Research from T. Mouats et al. [5] successfully implemented stereo thermal odometry on an unmanned aerial vehicle (UAV) for outdoor environments. Thermal features are obtained using the fast Hessian extractor and described with the FREAK descriptor. In their research, the authors added a comparison of different camera calibration methods. When evaluating different optimizers, it was discovered that the double dogleg algorithm provides better motion estimation with respect to other alternatives. The authors did not yet incorporate any trajectory optimization, such as loop closure detection. A global 3D map is not generated, but a local dense point cloud was created using a pair of stereo thermal images.

Borges et al. [6] utilized a monocular thermal camera and semi-dense optical flow estimation to determine the odometry of a road vehicle. The scale of the estimated path is subsequently obtained from inertial data and road plane segmentation. The researchers proactively triggered the camera’s Flat Field Correction (FFC) against Non-Uniform Noise (NUC) to prevent the loss of the image stream during critical moments such as turns. The optimal FFC triggering is determined according to temperature changes, the current vehicle turning rate, the upcoming turns and the time elapsed since the last correction. The method proposed by the authors is specifically tailored to the application of autonomous road vehicles. The algorithm namely relies on road plane segmentation to estimate the scale factor of the movement as well as a trajectory known a priori to trigger Flat Field Corrections.

The benefits of using the raw radiometric data from the thermal camera instead of rescaled 8-bit images were investigated by S. Khattak et al. [1]. The authors compared the performance of two monocular visual–inertial odometry systems operating on 8-bit rescaled thermal images, OKVIS and ROVIO, as well as two monocular thermal–inertial odometry algorithms, ROTIO and KTIO. The authors first explored the use of radiometric data by modifying ROVIO into ROTIO, which extracts FAST corner features in 8-bit rescaled images. The keypoints are then tracked across the frames in a semi-direct manner using the radiometric data. Improving on ROTIO, KTIO also detects features in the radiometric data according to their gradient value. The obtained points of interest are subsequently tracked in a semi-direct manner using the gradient image of the radiometric data. In contrast to rescaled 8-bit images, the authors noted that radiometric data are unaffected by contrast changes due to sudden temperature changes. Additionally, KTIO showed significantly higher operational robustness. The algorithm demonstrated accurate performance in situations where other visual–inertial and thermal–inertial odometry methods had already diverged during tests on a UAV in a dusty underground mine. Due to its absence in feature repeatability, the feature extraction used in KTIO is unsuitable for conventional feature-based loop closure detection methods, such as bag of visual words. The authors, therefore, recommend investigating more direct loop closure detection methods [7].

M.R.U. Saputra et al. [8] designed DeepTIO, a monocular thermal–inertial odometry algorithm utilizing a deep-learning approach. The optical flow from the thermal camera is estimated using the FlowNet-Simple network. In an improved version, an additional network incorporates "hallucinated" details into the thermal images. The hallucination network is trained from visual images from a multi-spectral dataset. The algorithm was tested, among other locations, in a smoke-filled room at a firefighters training facility. No fire source, however, was visible in the dataset. DeepTIO outperformed other thermal–inertial odometry methods, but the authors noted that the network lacks flexibility with respect to frame rate changes. Furthermore, it was observed that the hallucinated details did not provide any significant additional accuracy in the odometry estimation.

J. Jiang et al. [9] modified the visual deep-learning optical flow estimation algorithm RAFT for thermal images, ThermalRAFT. To decrease the image noise, the thermal images are pre-processed using a Singular Value Decomposition (SVD) that decomposes the image in independent linear components. The highest singular value is set to zero, dimming the image but also diminishing the noise. The remaining singular values are reallocated in ascending order, further diminishing the noise while also enhancing the details in low-contrast image regions. Gradient features similar to DSO are extracted from the thermal images before being tracked using the proposed ThermalRAFT algorithm. Potential loop closures are detected using the DBoW2 library by describing the extracted gradient features using the BRIEF descriptor.

Y. Wang et al. [10] and W. Chen et al. [11] extracted edge features using a Difference of Gaussians (DoG) filter instead of point features. The DoG filter’s bandpass filter behavior makes it more robust against image noise and low-contrast regions. Y. Wang et al. implemented the DoG edge filter in ETIO, a monocular thermal–inertial odometry algorithm for visually degraded environments that tracks the edge features using IMU-aided Kanade–Lucas–Tomasi feature tracking. ETIO was tested in outdoor settings, a mine dataset and an indoor environment filled with artificial smoke. Similarly, Chen et al. implemented the DoG edge filter into EIL-SLAM, a thermal–LiDAR edge-based SLAM algorithm. EIL-SLAM combines the estimated thermal and LiDAR odometry to increase the depth and scale estimation. A visual–thermal–LiDAR place recognition algorithm is utilized to detect loop closures for global bundle adjustment. The algorithm was tested in an urban scene during day- and nighttime.

With the exception of DeepTIO [8], all thermal odometry algorithms have only been tested in benign environments. Although the performance of DeepTIO [8] was evaluated in real smoke, no fire source was present in the vicinity. The effect of the fire source itself on the odometry estimation could, therefore, not be analyzed. Finally, a global trajectory optimization step, for example, in the form of bundle adjustment, has not been implemented in any of the current thermal odometry algorithms.

### 2.3. Thermal Mapping

S. Vidas et al. [12] developed a monocular SLAM system for hand-held thermal cameras. The algorithm detects corner features using the GFTT and FAST detectors and subsequently tracks them using sparse Lucas–Kanade optical flow. In order to process the thermal images, the raw 14-bit thermal data are converted to 8-bit from a fixed range centered around the mean value between the lowest and the highest 14-bit intensity. Matched SURF features are used in a homography motion estimation in order to resume tracking after an FFC. Only local trajectory optimizations are performed on all frames between two keyframes. A novel metric based on five criteria to select a new keyframe is additionally proposed. The authors claim that a reprojection error less than 1.5 can be considered as a good motion estimate. In most sequences, the proposed algorithm could not attain this value for longer periods. Being a semi-direct approach, the system notably failed during pure rotations. No information on the quality of the obtained 3D thermal map was presented by the authors.

Y. Shin et al. [13] combined depth data from a LiDAR and thermal data from a LWIR camera to create a thermal map. A sparse set of LiDAR points are projected into the 14-bit thermal image and are subsequently tracked in the following frame using a direct approach. Potential loop closures are detected using the bag of visual words approach by extracting ORB features from 8-bit images converted from a predefined temperature range. The authors note that this approach can result in a large amount of false positives in environments with a lack of detailed textures. An additional geometric verification is, therefore, presented to try to mitigate this phenomenon. The authors claim to have developed an all-day visual SLAM system as it is capable of computing an accurate trajectory and 3D thermal map irrespective of the outdoor lighting conditions.

E. Emilsson et al. [14] developed Chameleon, a stereo thermal–inertial SLAM system based on EKF-SLAM. The pose is estimated by tracking a maximum of thirty landmarks extracted using SIFT feature extraction and description. Experiments were conducted in both cold and heated environments, with the authors mentioning that positioning using only thermal–inertial data is not feasible in the former due to a lack of contrast in the thermal images. Although no quantitative error is provided, it is claimed that the algorithm estimates the position within a few meters inside a firefighting training facility where an active fire is present. The sparse mapping fails to provide a comprehensible map of the building. Finally, the authors note that, because of a temperature difference between the environment and the enclosed box in which the cameras are mounted, condensation was forming over time, leading to degraded tracking and mapping results.

When a thermal map is the only desired result, superimposing thermal data onto a 3D map generated from a SLAM algorithm utilizing a more common sensor is often preferred. Both the setup’s location and the depth data are then obtained from, for example, LiDAR measurements [15,16] or from an RGB camera [17,18] or from a depth camera [19]. Thermal information is then added to the 3D map by projecting the map points into the thermal images.

In benign environments, the construction of a thermal map is often aided by a second more reliable source of depth information. In a smoke-filled environment, these secondary sensors are, nevertheless, rendered useless. The thermal–inertial SLAM algorithm from E. Emilsson et al. [14], tested in a fiery environment, delivers a very limited map, and the system lacks global optimization capabilities. Finally, Chameleon suffers from lens condensation in fiery environments. Therefore, pose estimation heavily relied on inertial data for a significant part of the trajectory.

## 3. Materials and Methods

From the developments in thermal SLAM recently published, it becomes apparent that robust feature extraction and description remains one of the main challenges. Once a robust method to extract and describe thermal features is found, identical techniques to visual SLAM can be utilized to compute the odometry as well as generate the 3D map.

### 3.1. Setup and Datasets

Three thermal datasets were recorded during a fire incident inside the parking garage in Figure 3 at a firefighter training facility. Inside the building, two wood fires, visible in the datasets, were used as smoke sources and fueled between each recording. Along with the fire sources, a number of vehicles, wooden pallets and a shipping container are also located in the parking garage. The three datasets were captured, respectively, about five minutes, half an hour and an hour after ignition. The recordings were performed in the morning in early December when outside temperatures were just above freezing point.

The datasets were recorded using the setup in Figure 4 which includes two FLIR Boson 640 thermal cameras with a baseline of 80 millimeters. The thermal images are recorded at 30 frames per second and a resolution of 640×512. The stereo pairs are retrieved and synchronized according to the closest timestamp.

The active range of the 16-bit radiometric data is determined using its full cumulative histogram. The dynamic range is defined as $\left(p_{l},p_{u}\right]$, where $p_{l}$ is the defined lower bound of the cumulative percentage of 16-bit values and $p_{u}$ the defined upper boundary of the cumulative percentage of pixels to be included. The 8-bit value $I_{8}$ at the image location (u,v) is then the result of the linear transformation of the 16-bit intensity $I_{16}$:(1)$$I_{8}\left(u,v\right)=\frac{I_{16}\left(u,v\right)-I_{p_{l}}}{I_{p_{u}}-I_{p_{l}}}$$
where $I_{p_{l}}$ and $I_{p_{u}}$, respectively, correspond to the 16-bit intensity associated with $p_{l}$ and $p_{u}$. As significantly hotter objects compared to the environment are introduced into the frame, the dynamic conversion method above leads to a darkened scene. To avoid such a phenomenon, the upper intensity value $I_{p_{u}}$ is limited to a maximum value $I_{max}$. Finally, to further prevent sudden changes in brightness, the rescaled image is multiplied by a brightness gain determined by the ratio of a desired fixed brightness $b_{d}$ over the mean image brightness $\bar{b}$.

To rectify the thermal images, the cameras are calibrated using a plywood board containing an acircular calibration pattern. As shown in Figure 5, the top layer is heated using an infrared panel before being inserted inside the cold bottom layer. The intrinsic and extrinsic camera parameters are obtained using the camera calibration tool by J. Bowman et al. [20] included in the ROS *image_pipeline* package. The package utilizes the OpenCV calibration function based on Z. Zhang et al. [21] and J.-Y. Bouguet [22] while providing additional pre-processing functionalities for more robust results.

### 3.2. Feature Extraction and Description Benchmark

#### 3.2.1. Benchmarking Candidates

During the feature extraction and description benchmark, different combinations of extractors and descriptors are tested on thermal datasets recorded in a fiery environment at different temperatures and levels of smoke. As features are much more difficult to extract from thermal images due to their lower contrast, the default settings for the different candidates do not suffice. Instead, the different methods are tuned until a satisfactory amount of features are obtained within real-time computations. Table 1 presents the different feature extraction and description candidates included in the comparison. Different combinations of extractors and descriptors are tested in order to find the most suitable one to be used for thermal SLAM. For the classical extractors and descriptors (not marked with an "*" in Table 1), their implementation in OpenCV 4.5.0 [23] is used. The extracted features are matched using brute force, following the L2 norm for the floating-point descriptors SIFT, SURF and the CNN-based descriptor, whereas the Hamming distance is used for the binary descriptors ORB, BRIEF, BRISK and FREAK.

#### 3.2.2. Benchmark Criteria and Process

Following the benchmarks of feature extractors and descriptors on a thermal dataset by J. Johansson et al. [2] and by T. Mouats et al. [3], the performances of the different combinations of feature extractors and descriptors are evaluated based on the recall, i.e., the ratio of matches that are considered to be correct given the total amount of obtained matches, and the efficiency, i.e., the ratio of correct matches given the number of detected keypoints. A match is deemed correct if it is considered as an inlier by the RANSAC algorithm. Additionally, the number of detected keypoints is added as a criterion in this benchmark, which, along with the efficiency, provides the amount of points that can be utilized for the odometry and mapping calculations. As the destined application requires real-time performance, the computational speed of the combinations is also evaluated. The different benchmark criteria are summarized in Table 2. As the extractors do not necessarily provide the same number of features, it is more interesting to have matching metrics in the benchmark that are independent from the obtained number of features, such as rates. Finally, a metric capable of measuring the feature distribution across a frame is also included.

To measure the feature distribution across the frame, the image is divided into 320 equally sized bins. For each bin, the statistical divergence with respect to a uniform distribution across all bins using Pearson’s chi-squared metric [24] is computed. The entropy *H* of the entire frame is then obtained as the sum of Pearson’s chi-squared metric in each bin:(2)$$H=\sum_{i}{\left(\frac{P_{i}}{Q}-1\right)}^{2}\cdot Q$$
where $P_{i}$ is the observed feature distribution in the $i^{th}$ bin and *Q* is the theoretical uniform distribution. In Figure 6, an example of the thermal feature distribution for blobs and corners in an indoor benign environment is demonstrated. It can be noticed that the ORB corners are mostly located on the heat sources, i.e., the person and the ceiling lighting, resulting in high entropy. In contrast, the SURF blobs have lower entropy as the features are more dispersed across the scene.

The different major steps taken during the feature extractors and descriptors benchmark are summarized in Figure 7. Only the operations related to the feature detection and matching itself are timed. This means that the entropy computation is not included. Furthermore, the image rectification process is performed beforehand as it is identical for all candidates and is, therefore, also excluded from the computational time. With the exception of the CNN-based descriptors, the processing is performed on a laptop equipped with a quad-core Intel i7 processor at 1.8 GHz. The CNN-based descriptors are computed on a working station possessing an Intel Xeon Gold quad-core CPU at 3.6 GHz along with a NVIDIA Titan XP GPU.

#### 3.2.3. Gradient-Based Feature Tracking

Along with the classical feature detection and description algorithms, a gradient-based feature tracking algorithm based on KTIO [7] is included in the benchmark. For the initial testing, only a version capable of tracking features along the epipolar line in a stereo pair is implemented. In this method, features are extracted from a query image according to their gradient value. To assure a good distribution of the features across the frame, only features far enough apart from each other are added to the feature map. These features are then tracked in the second image, the tracking image, using template matching via normalized cross-correlation (NCC) on its gradient image. To limit the number of false positives, a tracked location is only valid if its NCC is above a given threshold. For comparison, gradient-based feature tracking is both implemented on raw 16-bit radiometric data as well as on the converted 8-bit thermal images. The gradient-based feature extraction in KTIO utilizes an image pyramid to extract and track features across two consecutive monocular frames. The different image scales are used to increase the tracking accuracy, which is initialized by inertial measurements [7]. Here, using stereo pairs, a feature’s location is more predictable as it lies on (or near) their corresponding epipolar line. A single image scale is, therefore, sufficient to characterize the viability of this feature tracking method. The full flowchart of this algorithm is depicted in Figure 8.

#### 3.2.4. CNN-Based Descriptors

With the increasing trend in recent years of utilizing machine learning for different image processing tasks, its suitability for matching features from thermal images is also assessed during this research. A Convolutional Neural Network (CNN) is trained to create a unique descriptor based on an image patch around the feature. The distance between a feature pair can then be measured using the L2 norm to determine the most suitable matches. Both the training and deployment are performed using Tensforflow 2.3 and Keras 2.4. The model utilizes the Siamese network, as shown in Figure 9, of two CNNs, which, in parallel, each compute a descriptor vector of an image patch centered around a feature using shared weights. Here, the CNN architecture from Y. Liu et al. [25] is adopted. Preliminary research, however, showed that increasing the size of the square input patch from 32 to 64 as well as increasing the descriptor size from 128 to 256 yield better results [26]. After having obtained the descriptors, the L2 norm between them is computed and scaled to a similarity score between 0 and 1 using a logistic function during training. The network is optimized using the contrastive loss function [27]. This loss function puts more of an emphasis on the predicted similarity measurement between the two input patches itself than more popular classification loss functions, such as binary cross-entropy. Given the true match label y={0,1} of the feature pair and the predicted similarity distance $0.0\leq \hat{y}\leq 1.0$, the contrastive loss *L* is defined as
(3)$$L\left(y,\hat{y}\right)=\frac{1}{2}\left(1-y\right){\hat{y}}^{2}+\frac{1}{2}\max{\left(1.0-\hat{y},0\right)}^{2}$$

The training dataset is composed of a total of 56,872 feature pairs divided equally over the match and non-match classes. For each class, 1620 pairs have been manually selected in stereo images from multiple datasets recorded during this work inside a laboratory and at the firefighting training facility. The remaining 26,816 pairs for each class have been generated using the KAIST Multispectral Pedestrian Detection Benchmark campus dataset [28], where features are detected and matched using ORB on monocular RGB–image pairs. The matched feature pairs are then projected onto the LWIR thermal images in order to extract the features’ region of interest.

### 3.3. FirebotSLAM

From the developments in thermal SLAM recently published, it becomes apparent that robust features extraction and description remains one of the main challenges. Once a robust method to extract and describe thermal features is found, identical techniques to visual SLAM can be utilized to compute the odometry as well as generate the 3D map. In Figure 10, a structural overview of FirebotSLAM is presented. Thermal features are extracted and matched across two consecutive frames as well as in the stereo pair. A state-of-the-art visual SLAM algorithm is then employed to perform the odometry, mapping and optimization tasks.

#### 3.3.1. Selection of the Slam Framework

The visual(–inertial) SLAM algorithm ORB SLAM 3 by C. Campos et al. [29] is adopted as the SLAM framework for FirebotSLAM. With respect to other state-of-the-art visual SLAM algorithms, ORB SLAM 3 offers greater robustness against loss of localization. The framework is not only able to utilize short-term information acquired within a few seconds but also long-term data gained since the start of the trajectory using a multi-map “atlas“ system. This multi-map storage method makes ORB SLAM 3 more resistant to localization failures compared to previous ORB SLAM iterations and other SLAM algorithms. In case the features failed to be tracked, a new map is created, which can be incorporated again into a previously obtained map inside the atlas when an overlap is detected at a later stage. This ability of long-term data association, together with its full reliability on Maximum-a-Posterior (MAP) estimation, ensured it was able to be two to five times more accurate than other state-of-the-art SLAM approaches on popular benchmarking datasets, such as EuRoC and TUM-VI [29].

Figure 11 displays the changes made inside the ORB SLAM 3 framework in order to accommodate the new feature extraction and description algorithms.

#### 3.3.2. Thermal Feature Detection and Matching

Features are extracted using the OpenCV 4.5.0 [23] implementation of the SURF extractor. Although both ORB and SURF provide scale invariance, SURF provides scale invariance by enlarging the filter size instead of shrinking the image size. As a result, the algorithm does not need to deal with the additional uncertainty (i.e., localization ambiguity) associated with downsampling the image as all operations are performed on a single image size. Limiting the feature matching between features of the same octave to reduce the number of outliers, however, remains. The optimal values providing sufficient thermal features within real-time computations for the different SURF parameters obtained after testing a large number of different value combinations on the recorded fire incident datasets proved to be a Hessian threshold of 5.0, a number of 4 octaves and 3 octave layers.

SURF features are described using the OpenCV 4.5.0 [23] implementation of the BRIEF descriptor. Because thermal images possess low contrast, more information about the features is beneficial to limit ambiguity. The SURF features extracted in FirebotSLAM are, therefore, described using the extended BRIEF descriptor of 64 bytes, in contrast to the 32 bytes of the ORB descriptor. To be able to detect potential loop closures using the DBoW2 bag of visual words library [30] with BRIEF, a vocabulary is generated from thermal features extracted from 1214 frames taken from a dataset recorded during a fire incident. Preliminary experiments of different tree shapes showed that adopting a vocabulary tree of six layers in depth and ten branches per layer allows BRIEF to recognize a sufficient number of potential loop closures [26]. A smaller tree lacks word diversity, leading to a greater number of false positives. Contrarily, a larger tree may describe frames too specifically, resulting in more false negatives. Experiments further revealed that reducing the cycle time between events where the optimizer searches for candidate frames to 1.25 s provides improved loop closure detections.

In addition to loop closure detection, the bag of words is also used as the primary feature matching method when tracking a local map. During map tracking, feature matching is namely limited between features that belong to the same node at a prescribed tree depth. Attempting to match features belonging to the same node in the fourth layer proves to provide the highest matching rate and recall. Lower levels increase the number of candidates and, subsequently, the risk for ambiguity, which lowers the recall. Matching features at higher levels decreases the number of candidates, resulting in a lower matching rate. During early testing, a high matching rate was obtained as well as a high recall from the ratio test. The majority of the matches were, however, rejected by RANSAC. Lightly incrementing the tolerated reprojection error, however, significantly increases the RANSAC recall to values similar as in the performed thermal benchmark [26]. A possible cause of this problem is the lower readability of blob features compared to corner features [3]. The location of certain identical features may then have slightly shifted in the scene across the frames during a change in viewpoint. Although their descriptors are still abundantly similar to be matched, the reprojection error between the two features is increased.

## 4. Results

### 4.1. Thermal Spectrum of the Datasets

A common risk of converting thermal radiometric data to 8-bit images is the sudden change in contrast due to temperature differences, possibly leading to a failure to track features. Figure 12 displays a typical 16-bit histogram over time of the datasets recorded at the firefighting training facility. In general, the environmental temperature, especially low to the ground, remains cool and well within the operating range of the thermal camera. During the first 50 s, as well as the interval of 90 to 110 s, the robot is facing the fire source. It can be noticed that the data are divided into two categories, environment and fire source. All the important information to perform thermal SLAM is contained within the first category, between 20,000 and 40,000. Therefore, the maximum intensity threshold $I_{max}$ for the range converted to 8-bit is set to 40,000. As shown in Figure 13, the information discarded above this threshold belongs to the fire sources, one inside the vehicle and one in the barrel next to the container, and the ceiling directly above them.

### 4.2. Feature Extraction and Description Benchmark

In Table 3, Table 4 and Table 5, the benchmark results for all the candidates are displayed. The results are standardized across all three datasets. The four possible color intensities for each criterion then indicate how near a candidate performs with respect to the absolute maximum (green) or minimum (red) value, depending on whether it performs above or below the general average. To keep the results clear, not all the results from all possible combinations of feature extractors and descriptors are included. For example, if a certain extractor provides a low amount of features, it is only displayed with its best-performing descriptor.

Looking at the feature extraction, it can be observed that blob feature extractors provide, although still plenty in the case of SURF, a smaller amount of features compared to corner extractors. On the other hand, the matchability, as well as the overall efficiency, of blob features is higher than corner features. This means that fewer blob features than corners need to be detected to obtain an identical number of correct and useful feature matches. Notably, SIFT does not provide a satisfying amount of features, which is consistent with the results from T. Mouats et al. [3]. Blob features offer slightly more robustness for motion estimation as they are more uniformly distributed across the frame than corner features. Corner features, on the other hand, by their nature, are more concentrated in image locations with a high image gradient. In thermal images, these locations correspond to areas of higher thermal gradient, such as close to the heat source and the ceiling. In visual images, a tradeoff between computational speed and recall needs to be made when using binary descriptors. This is not needed during this benchmark as binary descriptors, notably BRIEF and ORB (a rotation-invariant version of BRIEF), perform better than the floating-point descriptors at a faster rate.

The efficiency of a given descriptor appears to be dependent on its environment of application. The results from the three benchmarks, by Johansson et al. [2], T. Mouats et al. [3] and the one included in this paper, appear to differ. The top descriptor in Johansson’s et al. [2] benchmark is a floating-point descriptor, LIOP, whereas, in the remaining two, a binary descriptor was found to be the top contender. T. Mouats et al. [3] tested in benign environments and suggested the use of FREAK based on their findings, which performed as the worst descriptor in the benchmark performed here. The binary descriptors BRIEF and ORB showed to have a higher matching rate and recall compared to the floating-point descriptors SIFT and SURF. The latter compose their descriptors from the image gradient, which, in thermal images, has a lot less information compared to visual images due to the low contrast. Using binary comparison of pixel intensities appears to be a more robust method of describing thermal features. Additionally, the random testing pattern from (r)BRIEF is much more effective than the circular testing patterns from BRISK and FREAK.

Gradient-based feature tracking shows great potential for a very efficient method to detect and track features across thermal frames. This high efficiency can be explained by a more local search (partially) along the corresponding epipolar line. Its feature detection, however, provides no repeatability, making the features unsuitable for feature-based loop closure detection [7]. The lack of repeatability can be explained by the manner in which the gradient features are selected. Features are added to the feature map in an iterative manner from the top left to the bottom right of the gradient image. All features that are then too close to a previously selected feature are simply discarded as no metric is implemented to determine the best one. Using the tracking method with the radiometric data provides invaluable insight into the additional information and thus the performance the 16-bit data provides compared to the converted 8-bit data used by all visual-based algorithms. Furthermore, CNN-based descriptors present to be an interesting alternative to the classical descriptors, performing among the top contenders in the benchmark. For a given feature extractor, the CNN descriptors consistently offer a better feature distribution across the frame. The efficiency is, however, not groundbreaking, sometimes only slightly outperforming the best classical descriptor. Machine learning solutions are additionally notorious for their lack in flexibility. Therefore, the satisfying performance obtained during the benchmark is not guaranteed in all unknown situations. To generalize the algorithm for such flexibility, large amounts of data, time and effort (human and computational) are required [31]. Classical descriptors, on the other hand, already provide a degree of flexibility in different environments without any additional work required. Finally, from its mean frames per second, the current implementation of the CNN-based descriptors is not suitable for real-time odometry tasks.

Notably, between the first and second dataset, the performance of all candidates increased. As the temperature rises, the number of features extracted lowers, while the efficiency increases. The former is due to more locations in the building being masked out from extraction due to higher temperatures, causing data clipping by the thermal cameras. These regions are located close to the fire sources. The increase in temperature also results in more thermal gradients, thus providing more contrast to the thermal images. The additional contrast then helps to generate more distinctive descriptors, hence yielding a higher matching rate and recall.

A summary of the top candidates that emerged from the performed benchmark appears in Table 6. In spite of the generally impressive performance demonstrated by gradient-based feature tracking, its feature extraction lacks repeatability, resulting in features being inconsistently detected across varying viewpoints of an object. Consequently, this method is unsuitable for feature-based loop closure detection, a critical aspect within the chosen SLAM framework, ORB SLAM 3. CNN-based descriptors have emerged as a promising avenue for feature matching in thermal images. However, when compared to classical descriptors, their breakthrough potential is less apparent. Enhancing their capabilities could involve additional training utilizing blob features and a more diverse set of training data across various environments. A well-recognized issue with CNNs is their limited adaptability, implying that achieving high performance in one testing environment does not guarantee analogous results in a distinct location. Flexibility, a crucial attribute for firefighters navigating diverse and time-sensitive situations, remains challenging to attain, alongside the desired reliability. The sluggish processing speed associated with CNN-based descriptors might be mitigated through the adoption of more potent hardware, notably improved GPUs. On the other hand, the ORB feature extractor and descriptor yield a substantial number of keypoints and are tailored for real-time applications. However, their efficiency is somewhat constrained, necessitating the detection of a greater quantity of features to achieve a specific number of usable matches. This shortcoming is exacerbated by the distribution of keypoints across the image, negatively impacting both odometry estimation and the fidelity of the 3D map. In comparison, SURF-BRIEF and GFTT-BRIEF exhibit comparable performance. While GFTT demonstrates a slightly higher processing rate, SURF affords a superior distribution of features that remains consistently effective across the three TSC datasets. Notably, blob features are strategically positioned on surfaces, equipping them to provide more comprehensive environmental surface descriptions compared to corner-based features. Considering this attribute, the preference leans toward SURF-BRIEF over GFTT-BRIEF, positioning it as the prime candidate for integration within the ORB SLAM 3 framework.

The thin smoke generated by the wood fires did not affect the feature detection and matching process. In the coldest dataset, the smoke was only visible in the form of a light haze close to the ceiling. In the second and third datasets, the smoke had filled the entire room and was no longer visible on the thermal images.

### 4.3. FirebotSLAM

During the coldest recording, dataset 1, the obtained trajectory by the proposed algorithm, at first glance, strongly resembles the trajectory deduced by the thermal video. All objects present in the room are at least partly mapped by FirebotSLAM. The colder hood of the burning car located in the back right is partly located, as is the shipping container wall to the left of it. The exact placement of the container, however, possesses a large uncertainty, indicated by a large scatter of points. This side of the container is the most perceived element in the room as the robot navigates between the left and back side of the building. The scatter of points near the container wall is then the result of the accumulation of position errors over time. Some additional noise is also visible near the ground, where the height has been erroneously computed. Smaller objects are mapped in dataset 1 but do not contain enough detail to be recognized.

Despite that the loop closure between the vehicle entering and exiting the garage was detected, the same wall was mapped twice in different locations. The walls visible in the depth direction of the camera (left and back wall) are not created in the point cloud. This can mainly be attributed to the fact that the vehicle did not sufficiently approach these walls, leading to a lack of robust features. Additionally, the depth estimation error increases with the distance between the object and the camera, possibly leading to erroneous points on these walls. Dataset 1 was recorded a number of minutes after ignition. In the front left view of Figure 3, it can be noticed that the hot smoke, shown by the glowing haze, is located towards the ceiling. Although the smoke causes a haze, it does not seem to negatively affect the feature locations and, consequently, the odometry estimation. Instead, the hot smoke is heating the structure, leading to an increase in texture compared to objects lower to the ground.

The shortest recorded dataset, dataset 2 demonstrated in Figure 14, displays a rotated point cloud of the right wall with respect to the estimated path. These rotations only appear to happen to uniform surfaces visible in the left or right image border, where residual image distortion is more prominent as the vehicle drives alongside it. The correct orientation is, however, always obtained when the UGV faces a scene head-on. The pillar and scooter are mapped twice. During its first visit, the vehicle stops farther away from the location of interest and at a slightly different angle compared to the second visit. The bags of visual words from both frames are, therefore, too different, resulting in a failure to recognize the loop closure. The back, front and left wall of the building are here again not mapped. Part of the front left side of the building (including a car and a stack of pallets) is represented in the point cloud.

In the longest and hottest dataset, dataset 3, FirebotSLAM loses its position as it enters the garage. As, in this dataset, the robot does not create a loop closure when entering and exiting, the two maps in the atlas are never merged again. The heading of the walls computed by FirebotSLAM is erroneous in the front left part of the garage by about 10° after a loop closure optimization. In datasets 1 and 2, the robot exits the garage in reverse, whereas, in dataset 3, the robot rotates before facing the door as it exits. During this final turn, the heading error significantly increases. This large increase in error is mostly attributed to the lack of diverse features as only the door and wall are visible. Additionally, as it is the end of the dataset, no loop closure detection followed to correct the deviation.

As shown in Figure 15, FirebotSLAM has a mean reprojection error of 1.5 for all trajectories. The effect of the increasing amount of smoke across the datasets is not visible in the motion estimation. In contrast, the stock implementation of stereo ORB SLAM 3 running on thermal images provides a high reprojection error as well as greater uncertainty. Figure 14a further illustrates the differences in performance between both algorithms. While FirebotSLAM is able to keep the error to a minimum, the accumulation of errors by the visual SLAM algorithm ORB SLAM 3 results in a significant deviation in the end. As a result, ORB SLAM 3 fails to recognize many of the loop closures, including the final one, failing to improve the trajectory. The more uniform distribution of SURF features compared to ORB (also shown by the lower entropy of SURF in Table 3, Table 4 and Table 5) ensures that the 3D map obtained by Firebot SLAM has more detail with respect to the 3D map provided by the stock ORB SLAM 3, as shown in Figure 16.

For all three datasets, FirebotSLAM requires a total processing time of 120–130 ms per frame.

## 5. Discussion

When extracting thermal features, it was observed that blob features offer more matching robustness because they are distributed more uniformly across the frame. In contrast, corner features tend to be concentrated in locations with high image gradient, often corresponding to regions of high thermal gradient in thermal images. The lower amount of detected blob features is compensated by their higher matchability. In the end, this means that fewer blob features need to be detected than corners in order to obtain the same number of correct matches. The difference in recall between binary and floating-point descriptors is less apparent in thermal images compared to visual images. Binary descriptors generally outperformed floating-point descriptors in recall at a faster computation time. A possible explanation for this could be that binary descriptors are more robust to the high noise in thermal images.

The outcome of the benchmark results presented in Johansson et al. [2], in Mouats et al. [3] and in this article differ. In Johansson et al. [2], the top descriptor was selected to be LIOP, a floating-point descriptor, while, in the other two, a binary descriptor was selected. The recommended descriptor in Mouats et al. [3] proved to be one of the worst descriptors in the benchmark presented in this article. Both benchmarks, however, recommend SURF as the best thermal feature extractor.

Gradient-based features showed to be a very fast and efficient method for extracting and tracking features. Especially, the use of 16-bit radiometric data for tracking increased recall compared to other descriptors generated from 8-bit converted thermal images. Gradient-based features, however, provide no repeatability of location across frames. Because of this, it is incompatible for loop closure detection using visual bags of words. CNN-based descriptors showed great potential as a viable alternative to classical descriptors. Their increase in performance was, however, not groundbreaking, while their computation time is much longer. CNN-based descriptors also lack flexibility, meaning that the same efficiency is not guaranteed in different environments without further training.

The ORB feature extractor and descriptor provides a high number of keypoints and is designed for real-time applications. Its efficiency is, however, only sufficient, meaning that more features need to be detected to obtain a given amount of usable matches. Its keypoint distribution across the image is also not the best, which impacts both the odometry estimation as well as the detailing of the 3D map. SURF-BRIEF and GFTT-BRIEF exhibited similar performance. GFTT shows a slightly higher processing rate, whereas SURF provides a better feature distribution, which is also more consistent over the three datasets. Blob features are situated on surfaces, enabling them to better describe the environment’s surfaces in contrast to corners. Given this property, SURF-BRIEF is preferred over GFTT-BRIEF and will, therefore, be selected as the candidate to be implemented inside the ORB SLAM 3 SLAM framework.

Because FirebotSLAM utilizes ORB SLAM 3 as a SLAM framework, it also suffers from the same inconsistencies in performance. Running the same dataset through FirebotSLAM using exactly the same settings would not always yield the same results. Mainly, asynchronousity between the tracking and loop closure threads is observed. As a result, FirebotSLAM will sometimes miss important loop closures and consequently fail to track features in future frames or create a distorted map.

The synchronization of the stereo thermal pairs was performed digitally after acquisition by closest timestamp. It is, however, recommended to trigger both thermal cameras simultaneously using an external hardware connection as this guarantees that both clocks are synchronized as well as ensuring that the time difference between the capture of both cameras in a single pair is minimal.

Finally, FirebotSLAM was tested in a benign environment to perform a quantitative evaluation of its localization accuracy using the Motive Optitrack system. Despite having introduced different heat sources in the room, the algorithm failed to initialize as it failed to track a sufficient amount of features. Therefore, no meaningful results were obtained. In this benign environment at room temperature, noise was noticeably more visible in the thermal images than those recorded during the fire incident. The contrast in the images was also significantly lower in the benign environment, which resulted in much more ambiguity in the features’ descriptors.

## 6. Conclusions

Despite the notable challenges associated with thermal cameras, the first SLAM algorithm capable of creating a coherent 3D map solely from thermal images in a fiery environment was demonstrated in this article. The main challenge of thermal SLAM is the detection of robust features. Therefore, a practical benchmark of different feature extraction and description methods was performed on a thermal datasets record during a fire incident. The selected combination of extractor and descriptor is then implemented into the state-of-the-art visual SLAM algorithm ORB SLAM 3 to form FirebotSLAM. The performance of the presented algorithm is then compared against the stock ORB SLAM 3 algorithm in a fire incident.

During the benchmark, it was observed that blob features offer better feature distribution across the frame, whereas the extraction time is faster for corner features. Additionally, the BRIEF binary descriptor performed competitively with respect to floating-point descriptors, such as SURF and the CNN-based descriptor, at a much faster processing rate. Gradient-based feature tracking provides a very fast and efficient method to track features given an initial good estimate of location. Due to its nature, however, it is not suited for feature-based loop closure methods, so alternatives for loop closure detection would need to be investigated. CNN-based descriptors showed great results, often slightly outperforming classical descriptors in terms of recall and feature distribution. They, unfortunately, operate very slowly and are, therefore, not suited for real-time SLAM. To conclude, the SURF extractor combined with the BRIEF descriptor proved to be the optimal combination to obtain robust features for real-time thermal SLAM during a fire incident.

By implementing the SURF-BRIEF feature extractor and descriptor in ORB SLAM 3, it is possible to localize a vehicle and create a 3D map from stereo thermal images in a smoke-filled environment during a fire incident. The FirebotSLAM system can estimate the robot’s trajectory in real time and generate a consistent 3D map by detecting important loop closures. Using a stereo thermal setup instead of a monocular thermal camera allows for obtaining the trajectory and 3D map at the correct scale without relying on an Inertial Measurement Unit. While no ground truth was available, the obtained odometry is visually consistent and precise enough for path reversal and obstacle avoidance. The 3D map includes the location of obstacles, including large objects not known from prior knowledge. The use of SURF features results in a more meaningful 3D map compared to stock ORB SLAM 3. The FirebotSLAM system increases situational awareness for both the operator and mission planning, which was confirmed by firefighters involved in the project.

The smoke present inside the environment did not seem to cause any significant averse effects on the feature extraction and matching and thus the odometry estimation. It could even be argued that it proved to be beneficial to the successful operation of FirebotSLAM. It was namely observed that the heat provided by both the fire source itself and by the emitted smoke increased the differences in measured infrared emissivity, resulting in more contrast in the thermal images. It can be concluded that performing thermal SLAM in a smoke-filled building in the presence of fire is equivalent to performing visual SLAM at night with a giant floodlight turned on.

In future work, a quantitative measurement of the odometry accuracy of FirebotSLAM will be performed. Comparing FirebotSLAM with other state-of-the-art thermal SLAM algorithms is also desirable. Furthermore, the fusion of inertial and thermal odometry in FirebotSLAM will be researched as to further increase the algorithm’s localization accuracy and robustness, especially when insufficient robust features are extracted. Additionally, stereo dense reconstruction will be investigated in order to create dense point clouds from thermal images. The benefits of using the 16-bit raw radiometric data both for feature extraction and matching as well as for dense reconstruction will also be examined. Raw radiometric data could namely offer additional robustness in environments farther away from a fire source, where thermal image contrast is expected to be lower. Finally, the performance of FirebotSLAM in thicker smoke, e.g., commonly produced by burning composites in modern cars, should be investigated.

## Figures and Tables

**Figure 1 sensors-23-07611-f001:**
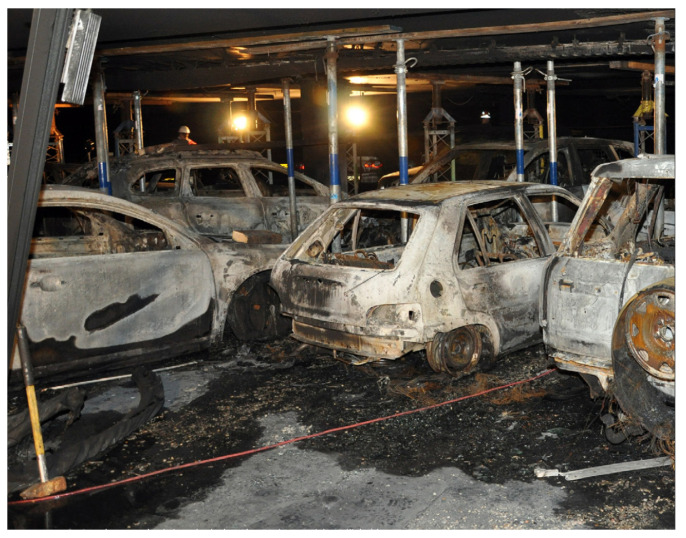
Fire damage in the parking garage De Appelaar in Haarlem, The Netherlands (credit: Gemeente Haarlem).

**Figure 2 sensors-23-07611-f002:**
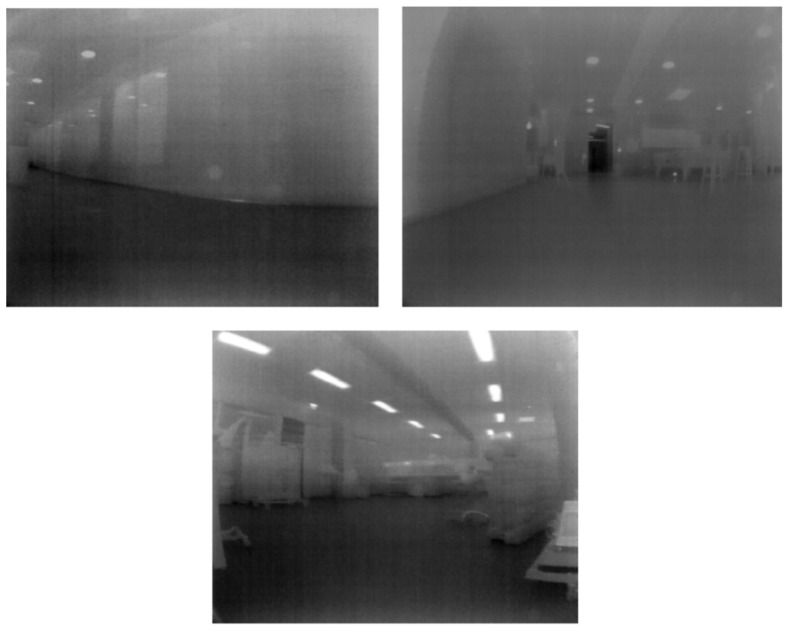
Thermal images recorded inside a benign indoor environment.

**Figure 3 sensors-23-07611-f003:**
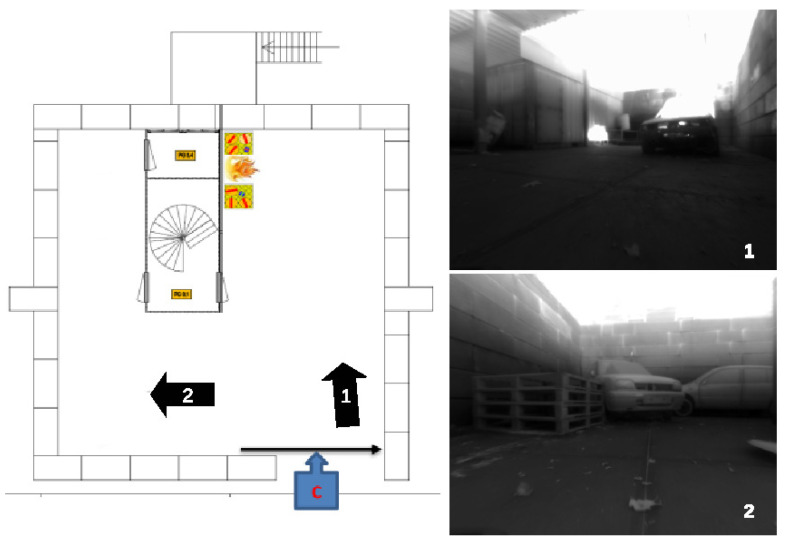
Layout of the parking garage. The two images on the right have been taken at the pose of the corresponding arrow in the map on the left.

**Figure 4 sensors-23-07611-f004:**
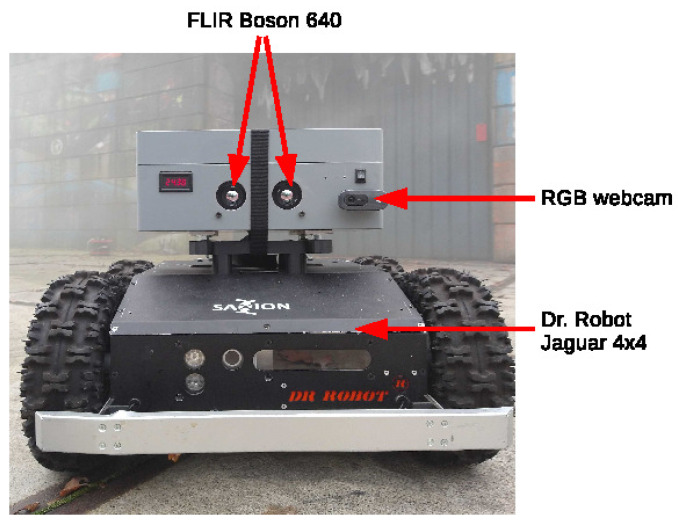
Firebot measurement setup.

**Figure 5 sensors-23-07611-f005:**
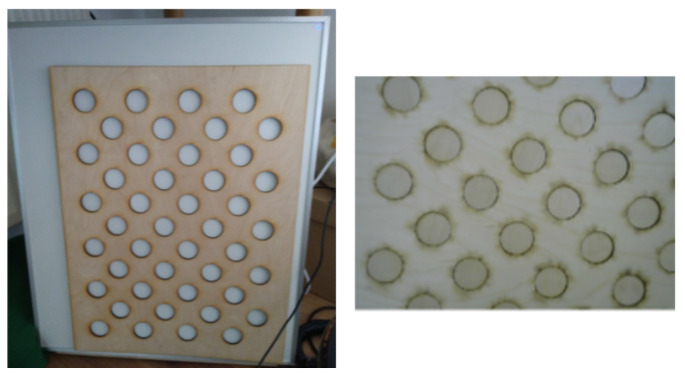
Plywood thermal calibration board.

**Figure 6 sensors-23-07611-f006:**
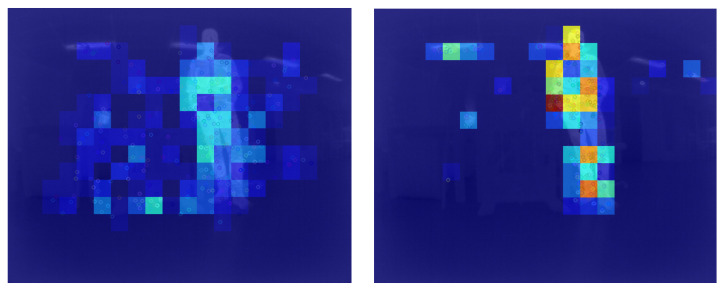
Distribution of correctly matched SURF (**left**) and ORB (**right**) thermal features in a benign indoor environment. A dark red bin signifies a high feature density, whereas a dark blue bin does not contain any features.

**Figure 7 sensors-23-07611-f007:**
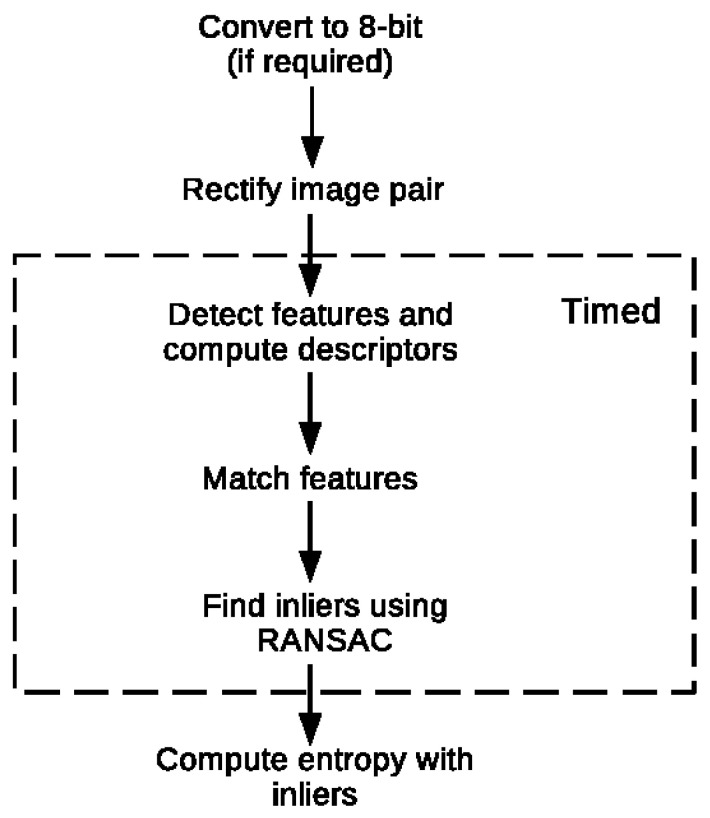
Benchmarking process.

**Figure 8 sensors-23-07611-f008:**
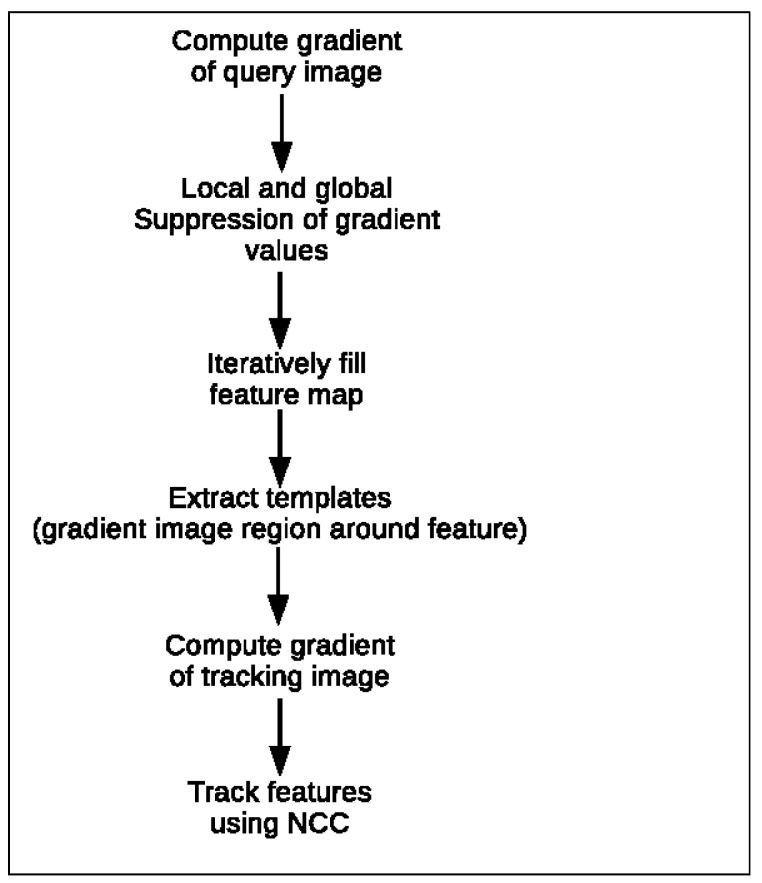
Gradient-based features flowchart.

**Figure 9 sensors-23-07611-f009:**
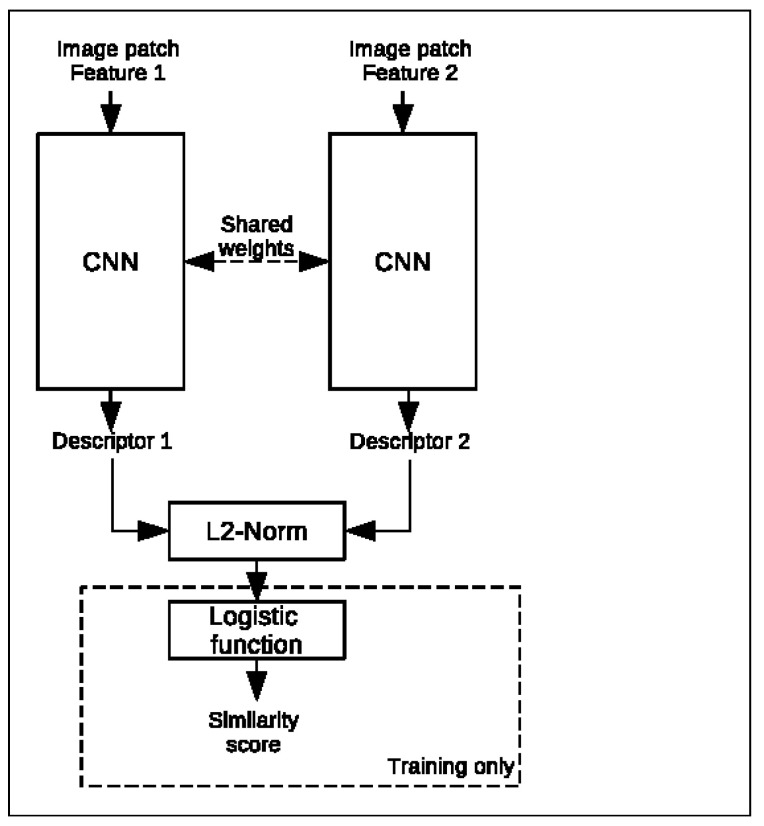
Siamese CNN architecture for feature matching.

**Figure 10 sensors-23-07611-f010:**
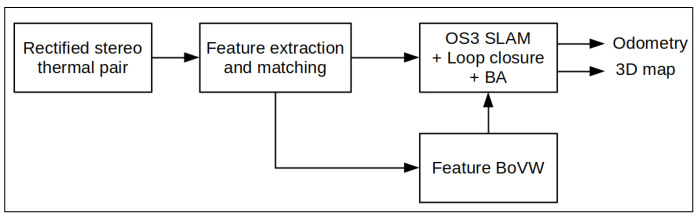
Firebot thermal SLAM structural overview. Features are extracted and matched from a pair of rectified stereo thermal images to compute the 3D points and estimate the pose. Feature vectors from a bag of visual words (BoVW) are also created from the images to enhance feature matching as well as to detect potential loop closures. The robust thermal features are then fed to the optimizers of the state-of-the-art visual SLAM algorithm ORB SLAM 3 (OS3) to compute the odometry and perform the global bundle adjustment (BA).

**Figure 11 sensors-23-07611-f011:**
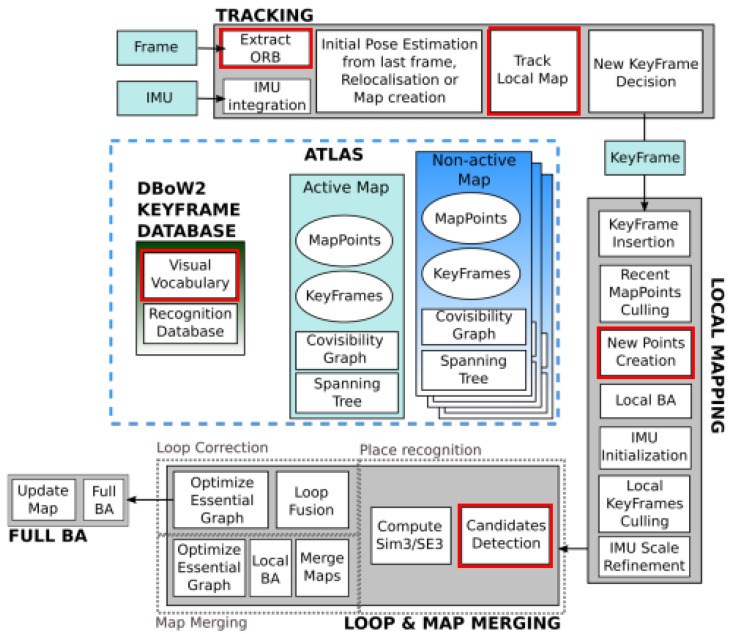
ORB SLAM 3 architecture diagram [29]. The elements that have been modified for FirebotSLAM are highlighted by the red bounding boxes.

**Figure 12 sensors-23-07611-f012:**
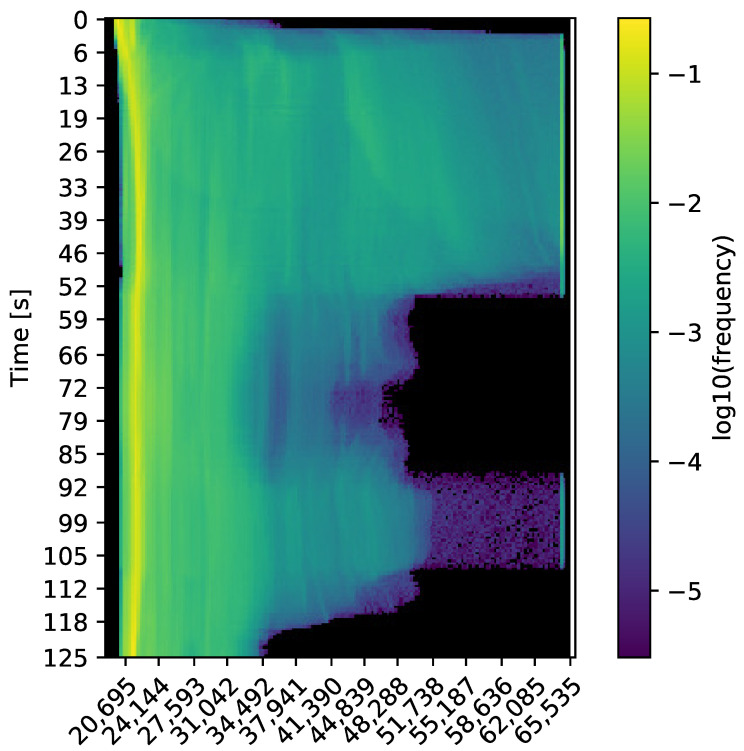
Typical 16-bit thermal histogram over time of the acquired datasets during a fire incident.

**Figure 13 sensors-23-07611-f013:**
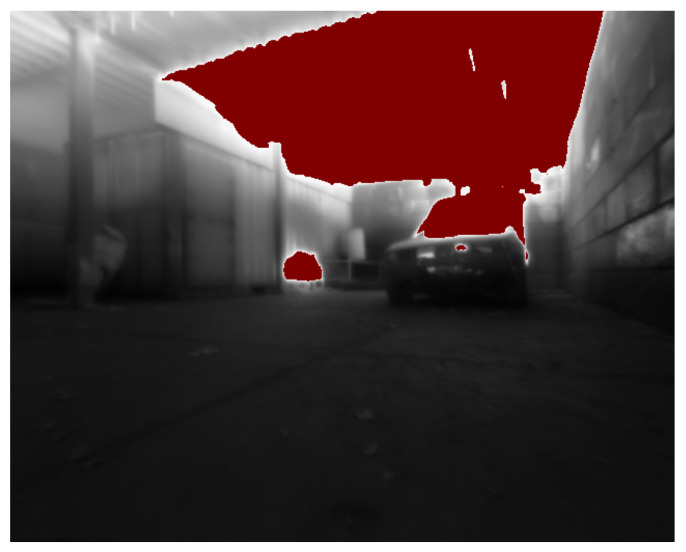
Red mask showing elements of the scene above 40,000 when facing both fire sources.

**Figure 14 sensors-23-07611-f014:**
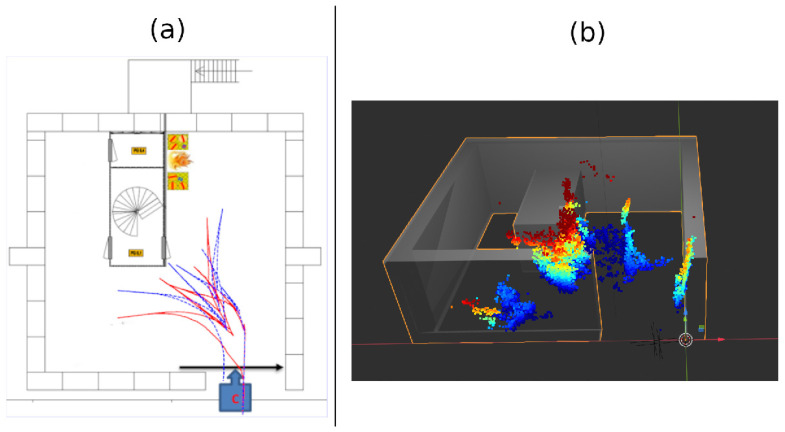
(**a**) Computed path of dataset 2 by FirebotSLAM (red) and ORB SLAM 3 (blue). (**b**) Pointcloud of dataset 2 generated by FirebotSLAM. For reference, a transparent CAD model of the garage’s wall layout has been added.

**Figure 15 sensors-23-07611-f015:**
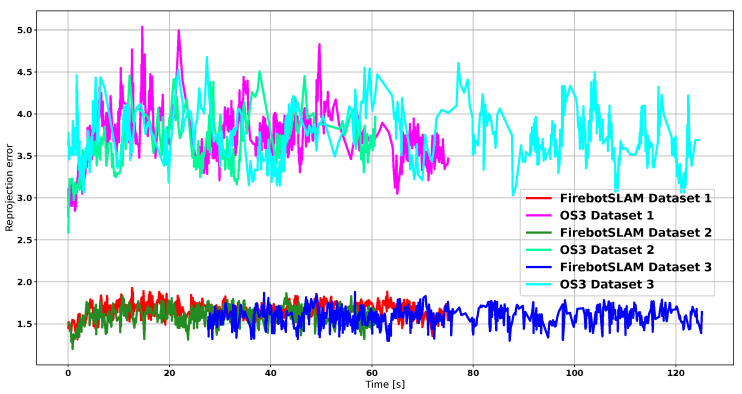
Reprojection error by FirebotSLAM and OS3 across the three datasets. The first 25 s of dataset 3 were lost by FirebotSLAM due to a tracking failure.

**Figure 16 sensors-23-07611-f016:**
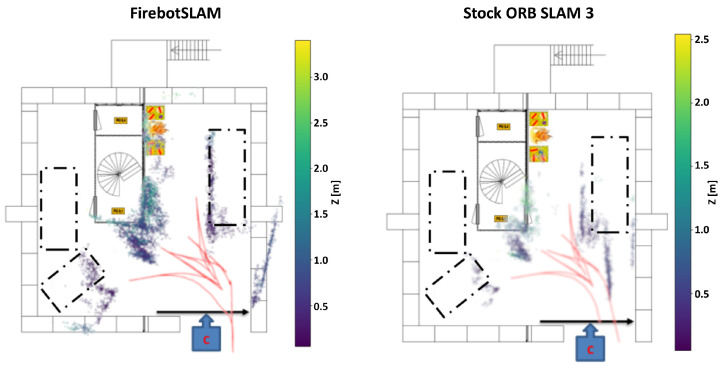
Comparison between Firebot SLAM and the stock ORB SLAM 3 of the generated map of dataset 2.

**Table 1 sensors-23-07611-t001:** Benchmark candidates. The candidates not present in OpenCV are marked with an “*”.

Extractors		Descriptors	
SIFT	SURF	SIFT	SURF
FAST	ORB	BRIEF	ORB
GFTT	Gradient-based *	BRISK	FREAK
		CNN-based *	

**Table 2 sensors-23-07611-t002:** Benchmark criteria.

Criterion	Description
Mean number of keypoints	Average number of
	keypoints detected across the frames
Mean matching ratio	Average percentage of matched
	keypoints in the left image
Mean recall	Mean percent of matches considered
	as correct by RANSAC outlier removal
Mean efficiency	Mean percent correct matches
	for a given number of detected keypoints
Mean feature	Metric to determine the distribution
distribution entropy	of features across the frame
Mean frames per second	Average frames processed in a second

**Table 3 sensors-23-07611-t003:** Benchmark results from dataset 1 (the coldest). The green or red intensity of each cell indicates, respectively, how much the candidate performs either above or below the general average across all datasets. The best candidates are highlighted in bold.

Name	FPS	Keypoints	Matching Ratio	Recall	Efficiency	Entropy
SIFT	8.4	580	0.5328	0.6699	0.3569	7723
SURF	4.6	1241	0.4795	0.6924	0.332	3316
**SURF-BRIEF**	7.3	1241	0.5101	0.8373	0.4271	3129
SURF-FREAK	8.6	1241	0.3932	0.7766	0.3054	4075
SURF-BRISK	8.1	1241	0.4279	0.823	0.3521	3604
SURF-ORB	11.5	1241	0.4456	0.7758	0.3457	3613
SURF-CNN_64_256	0.6	1241	0.5608	0.7356	0.4126	2681
ORB-BRIEF	12.3	1795	0.3632	0.8865	0.322	8333
ORB-FREAK	30.4	1795	0.176	0.8449	0.1487	17007
ORB-BRISK	19.3	1795	0.263	0.8369	0.2201	13451
ORB	15.2	1795	0.4446	0.7732	0.3437	8961
**ORB-CNN_64_256**	0.4	1795	0.3994	0.8898	0.3554	7339
Fast-BRIEF	12.6	1789	0.4466	0.7647	0.3415	5684
GFTT (Shi-Tomasi)-SURF	10.2	1484	0.343	0.057	0.0195	26575
**GFTT (Shi-Tomasi)–BRIEF**	16.6	1484	0.4825	0.8464	0.4084	6524
GFTT (Shi-Tomasi)–FREAK	16.4	1484	0.3619	0.6313	0.2284	7700
GFTT (Shi-Tomasi)–BRISK	12.3	1484	0.374	0.7477	0.2796	7313
GFTT (Shi-Tomasi)–ORB	16.6	1484	0.4805	0.8233	0.3956	6306
**Gradient-based (8-bit)**	43.0	955	0.8346	0.8469	0.7068	939
**Gradient-based (16-bit)**	44.0	950	0.9453	0.8586	0.8116	579

**Table 4 sensors-23-07611-t004:** Benchmark results from dataset 2. The green or red intensity of each cell indicates, respectively, how much the candidate performs either above or below the general average across all datasets. The best candidates are highlighted in bold.

Name	FPS	Keypoints	Matching Ratio	Recall	Efficiency	Entropy
SIFT	9.2	521	0.5816	0.7624	0.4434	6111
SURF	5.6	1173	0.5754	0.8163	0.4697	2164
**SURF-BRIEF**	7.3	1173	0.6104	0.8966	0.5473	2142
SURF-FREAK	8.3	1173	0.4817	0.8637	0.416	2784
SURF-BRISK	7.9	1173	0.5354	0.8838	0.4731	2443
SURF-ORB	11.0	1173	0.5644	0.861	0.4859	2359
**SURF-CNN_64_256**	0.6	1173	0.6436	0.8848	0.5695	1953
ORB-BRIEF	13.1	1700	0.4	0.9029	0.3612	6788
ORB-FREAK	31.5	1700	0.1918	0.8712	0.1671	15046
ORB-BRISK	20.3	1700	0.2847	0.8554	0.2435	11499
ORB	15.2	1700	0.4988	0.8113	0.4047	7247
**ORB-CNN_64_256**	0.5	1700	0.4359	0.9177	0.4	6051
**Fast-BRIEF**	25.8	923	0.5623	0.8439	0.4745	5946
GFTT (Shi-Tomasi)–SURF	12.4	1433	0.3482	0.0842	0.0293	17925
**GFTT (Shi-Tomasi)–BRIEF**	13.9	1433	0.5785	0.8999	0.5206	4442
GFTT (Shi-Tomasi)–FREAK	17.3	1433	0.441	0.7563	0.3336	5013
GFTT (Shi-Tomasi)–BRISK	12.3	1433	0.4976	0.8457	0.4208	4513
**GFTT (Shi-Tomasi)–ORB**	16.4	1433	0.5729	0.8843	0.5066	4251
**Gradient-based (8-bit)**	43.9	909	0.813	0.8457	0.6876	805
**Gradient-based (16-bit)**	37.0	1085	0.9244	0.8514	0.7871	395

**Table 5 sensors-23-07611-t005:** Benchmark results from dataset 3 (the hottest). The green or red intensity of each cell indicates, respectively, how much the candidate performs either above or below the general average across all datasets. The best candidates are highlighted in bold.

Name	FPS	Keypoints	Matching Ratio	Recall	Efficiency	Entropy
SIFT	9.7	479	0.595	0.7719	0.4593	11714
SURF	5.2	1177	0.5862	0.8261	0.4843	4325
**SURF-BRIEF**	7.7	1177	0.6245	0.9007	0.5624	4341
SURF-FREAK	9.3	1177	0.4928	0.8741	0.4308	5589
SURF-BRISK	7.9	1177	0.5548	0.8913	0.4945	4904
SURF-ORB	11.3	1177	0.5854	0.8679	0.5081	4699
**SURF-CNN_64_256**	0.6	1177	0.6491	0.8927	0.5794	3924
ORB-BRIEF	12.9	1692	0.4025	0.9075	0.3652	12645
ORB-FREAK	33.2	1692	0.1684	0.8772	0.1478	31607
ORB-BRISK	21.5	1692	0.26	0.8591	0.2234	22690
ORB	15.2	1692	0.5053	0.8199	0.4143	13290
ORB-CNN_64_256	0.5	1692	0.4403	0.9195	0.4048	11418
Fast-BRIEF	29.0	844	0.5652	0.847	0.4787	11164
GFTT (Shi-Tomasi)–SURF	11.9	1492	0.3458	0.0911	0.0315	34021
**GFTT (Shi-Tomasi)–BRIEF**	13.3	1492	0.5891	0.9033	0.5322	8266
GFTT (Shi-Tomasi)–FREAK	15.3	1492	0.4517	0.7641	0.3452	9350
GFTT (Shi-Tomasi)–BRISK	12.7	1492	0.5127	0.8497	0.4357	8377
**GFTT (Shi-Tomasi)–ORB**	16.9	1492	0.5871	0.8893	0.5221	7888
**Gradient-based (8-bit)**	41.2	946	0.8087	0.8458	0.6839	1466
**Gradient-based (16-bit)**	35.8	1089	0.9256	0.8522	0.7888	797

**Table 6 sensors-23-07611-t006:** Comparison of the top candidates.

Candidate	Number of Keyppoints	Efficiency	Distribution of Keypoints	Real Time	BoVW Compatible
SURF-BRIEF	Good	High	Very good	Sufficient	Yes
SURF-CNN	Good	High	Very good	No	Yes
ORB	High	Sufficient	Sufficient	Very good	Yes
ORB-CNN	High	Sufficient	Good	No	Yes
GFTT (Shi-Tomasi)-BRIEF	Good	Very good	Good	Good	Yes
Gradient-based (16-bit)	Sufficient	Excellent	Excellent	Excellent	No

## Data Availability

Datasets used in the experiments are available on request.

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
