# Peer review of "FirebotSLAM: Thermal SLAM to Increase Situational Awareness in Smoke-Filled Environments"

_sensors, 2023, doi:10.3390/s23177611_

Round 1
Reviewer 1 Report
This paper presents a method for visual SLAM using a thermal camera and a ground vehicle for deploying this method. The paper is well structured, contains adequate background information, and is thoroughly validated. However, the following comments are suggested to improve the overall quality of the paper.
Line 54: this second bullet is not true. There was a paper in 2018 that proposed and validated thermal visual mapping:
Real-time Energy Audit of Built Environments: Simultaneous Localization and Thermal Mapping, Ramachandra, B., Monroe, J., Nawathe, P., Han, K., Ham, Y., and Raju Vatsavai, R. Journal of Infrastructure Systems, ASCE, 24(3), 2018.
There was another paper that maps thermal image (colors from normalized temp) to 3D point cloud resulting from vision/image-based 3D reconstruction that the authors should look at:
An Automated Vision-based Method for Rapid 3D Energy Performance Modeling of Existing Buildings using Thermal and Digital Imagery, Ham, Y., and Golparvar-Fard, M. Advanced Engineering Informatics, Elsevier, 27(3), 395-409, 2013.
Line 411-412: what do you mean by uncertainty with downsampling of images? On the other hand, there is no uncertainty with scale invariant by filter sizes? Please clarify/add more details
Line 417: explain "extensive testing" to find threshold, # of octave and layers. Do these parameters need to be changed when you have a different scene and camera with different lenses and sensor sizes? Or can you use the same number for any thermal SLAM?
Author Response
Dear reviewer,
Thank you for taking the time to read our manuscript.
Regarding the notes and suggestions made:
- The second bullet point in Line 54 has been adapted to be limited to the scenario of a fire incident. Because the authors could not get access to the suggested paper by B. Ramachandra et al., they decided to not cite the paper without reading it.
- A number of additional and more recent references have been added to the "related work" section, including the suggested article by Y. Ham et al.
- The sentences in line 411-412 have been rephrased and clarified. Hopefully this will clear any further confusion.
- "Extensive testing" has been clarified as to have tested a large number of different combinations of parameter values, on the recorded fire incident datasets. The selected parameters have not been tested with other sensors, lenses or in different scenes.
Reviewer 2 Report
1. How does FirebotSLAM, as the first algorithm of its kind, contribute to the enhancement of situational awareness in fire incidents, particularly in cases with high levels of smoke?
2. Can you explain how the combination of extractor and descriptor contributes to FirebotSLAM's functionality, and how this differs from other algorithms like ORB SLAM 3?
3. How has FirebotSLAM demonstrated superior performance in generating a meaningful 3D map and detecting important loop closures compared to the stock ORB SLAM 3 algorithm?
4. Could you elaborate on the reasons behind using SURF features in FirebotSLAM and how it contributes to generating a more meaningful 3D map than ORB SLAM 3?
5. How does FirebotSLAM augment the operational efficiency and situational awareness for operators and mission planning, based on the feedback from firefighters involved in the project?
6. Considering the current focus of research on thermal odometry over thermal SLAM, how does the FirebotSLAM algorithm contribute to this field of research, and what potential future implications can you foresee?
7. Can you explain the process used for thermal mapping and the utilization of thermal images overlaid onto a point cloud obtained from another depth source in the FirebotSLAM algorithm?
8. Could you discuss the limitations or challenges in the way data was collected for the study, particularly the timing, outdoor conditions, and thermal datasets' active range?
9. While FirebotSLAM performed well in a single fire incident scenario, how might its performance vary in other fire incidents or different environments?
10. Can you shed light on why FirebotSLAM's performance was not successful in a benign environment and whether this indicates the algorithm's potential limitations outside of fire scenarios?
11. How does the lack of diverse features in some datasets impact the algorithm's overall accuracy and what measures could be taken to mitigate the increase in heading error?
12. Given that FirebotSLAM has not been compared to other thermal SLAM algorithms, how can we assess its performance and effectiveness? Do you have plans to compare it with other state-of-the-art methods in future research?
Author Response
Dear reviewer,
Thank you for taking the time to read our manuscript.
Regarding the notes and suggestions made:
- Details have been added at the end of the introduction explaining what benefits the partnering firefighters saw in FirebotSLAM (lines 62-67).
- Figure 15 showing a top-down view of the point density of the maps obtained by FirebotSLAM and the stock ORB SLAM 3 have been added. Information about the difference in results is provided in the final paragraph the "results" section (lines 666-669).
- See 2.
- A more detailed explanation of why SURF-BRIEF was selected as the extractor and descriptor for the SLAM framework and Table 6 have been added in the "results" section (lines 573-602).
- See 1.
- I am sorry but this question is not clear to the authors. The contributions of the research to the field are provided in the introduction while future challenges and topics that will be handled are described in the final paragraph of the conclusion.
- No other depth source was used in this research. The depth information in FirebotSLAM was solely obtained from a stereo pair of thermal cameras.
- A comment has been added regarding camera time synchronization in the discussion (lines 722-726). The outdoor conditions did not impact the data acquisition performed indoors. The environment itself remained relatively cool and well within the operating range of the thermal cameras. This remark has been added to section 4.1 regarding the thermal spectrum analysis.
- No conclusions can be made on FirebotSLAM's performance in other (fire incident) scenarios as it has not been tested elsewhere. As mentioned in the conclusion, testing FirebotSLAM in thicker smoke is one of the future works listed. Regarding other environments in general, this is out-of-scope (for the exception of the ) for the moment as FirebotSLAM was designed with firefighters for firefighters to use during fire incidents only.
- Additional information has been added in the discussion (lines ).
- The points mentioned in the future work section related to improving the feature extraction and tracking are related to increasing the robustness of FirebotSLAM in conditions of low features. Additionally, including IMU data in the motion estimation will also improve FirebotSLAM's accuracy, notably when estimating the heading.
- Yes, we plan to compare FirebotSLAM with other SOTA thermal SLAM algorithms when they are available publicly. This has been added in the future works section. Unfortunately, during the research period, we could not find access any code repositories of the algorithms mentioned in the related work section.